# Early Ustekinumab Use Improves Clinical Outcomes in Biologic-Naive Crohn’s Disease Patients: A Retrospective Multicenter Cohort Study in Taiwan

**DOI:** 10.3390/biomedicines13020391

**Published:** 2025-02-06

**Authors:** Yen-Cheng Chang, Chiao-Hsiung Chuang, Tien-Yu Huang, Chen-Shuan Chung, Chia-Jung Kuo, Yu-Bin Pan, Puo-Hsien Le

**Affiliations:** 1School of Medicine, Chang Gung University, Taoyuan 333, Taiwan; b0902016@cgu.edu.tw; 2Department of Internal Medicine, National Cheng Kung University Hospital, College of Medicine, National Cheng Kung University, Tainan 701, Taiwan; jasonc@mail.ncku.edu.tw; 3Taiwan Association for the Study of Intestinal Diseases (TASID), Taoyuan 333, Taiwan; tienyu27@gmail.com (T.-Y.H.); chungchenshuan_3@yahoo.com.tw (C.-S.C.); m7011@cgmh.org.tw (C.-J.K.); 4Division of Gastroenterology, Tri-Service General Hospital, National Defense Medical Center, Taipei 114, Taiwan; 5Division of Gastroenterology and Hepatology, Department of Internal Medicine, Far Eastern Memorial Hospital, New Taipei City 220, Taiwan; 6Department of Gastroenterology and Hepatology, Chang Gung Memorial Hospital at Linkou, Taoyuan 333, Taiwan; 7Chang Gung Inflammatory Bowel Disease Center, Chang Gung Memorial Hospital at Linkou, Taoyuan 333, Taiwan; 8Biostatistical Section, Clinical Trial Center, Chang Gung Memorial Hospital at Linkou, Taoyuan 333, Taiwan; e8901145@gmail.com; 9Chang Gung Microbiota Therapy Center, Chang Gung Memorial Hospital at Linkou, Taoyuan 333, Taiwan

**Keywords:** early biologics, ustekinumab, Crohn’s disease, CDAI, clinical outcomes

## Abstract

**Background/Objectives**: Crohn’s disease (CD) is a progressive condition, and early treatment with infliximab combined with an immunosuppressant within six months has been shown to improve clinical outcomes. However, the impact of early ustekinumab (UST) use in biologic-naïve CD patients remains unclear. This study aims to address this gap by evaluating the clinical outcomes of early UST intervention in such patients. **Methods**: In this retrospective cohort study, we included biologic-naïve CD patients treated with UST, with a clinical follow-up period of at least six months from October 2020 to January 2024 across four medical centers. Patients who received UST within six months of CD diagnosis were categorized into the Early-UST group, while those who were initially treated with conventional therapies and subsequently received UST after six months were assigned to the control group. The primary endpoint was the improvement of clinical outcomes at six months. **Results**: A total of 60 biologic-naïve CD patients were enrolled. Baseline characteristics were comparable between the two groups. At six months, the Early-UST group (n = 24) demonstrated significantly lower Crohn’s Disease Activity Index (CDAI) scores (73.03 vs. 112.42, *p* = 0.038), lower Harvey–Bradshaw Index (HBI) scores (1.46 ± 1.69 vs. 2.72 ± 2.17, *p* = 0.020), higher rates of clinical remission (91.7% vs. 63.9%, *p* = 0.017), and higher rates of steroid-free clinical remission (79.2% vs. 50.0%, *p* = 0.031) compared to the control group (n = 36). At one year, the early-UST group continued to exhibit lower CDAI scores (39.94 vs. 91.48, *p* = 0.005). **Conclusions**: Initiating ustekinumab within six months of CD diagnosis is associated with improved clinical outcomes and enhanced quality of life in biologic-naïve Crohn’s disease patients.

## 1. Introduction

Crohn’s disease (CD) is a chronic, progressive disorder involving the gastrointestinal tract that can result in complications such as fistulas, abscesses, and strictures [1]. Since the introduction of infliximab in CD in 1998, an increasing number of biologics with distinct mechanisms have become available for the treatment of CD. The advent of biologic therapies has significantly reduced surgical risk for CD patients [2]. However, despite these improvements, there are still some patients who require surgical intervention [3]. The timing of biologic therapies may also play a crucial role in modifying the disease course and improving the clinical outcomes. Studies have shown that early biologic treatment within one to three years of CD diagnosis is associated with lower surgery rates and higher remission rates compared to delayed treatment [4,5,6,7,8]. These studies primarily focused on the impact of anti-tumor necrosis factor alpha (anti-TNF-alpha), demonstrating that early use of biologics may lead to superior clinical outcomes for patients with CD.

Building on these findings, the ECCO guidelines discourage the use of 5-aminosalicylates (5-ASA) and thiopurine monotherapy as induction therapy for CD, as well as the use of 5-ASA and steroids for maintaining remission in CD [9]. The American Gastroenterological Association (AGA) similarly advocates for the early introduction of biologics, with or without an immunomodulator, rather than delaying their use until after the failure of 5-ASA and/or corticosteroids [10]. The Paris definition suggests that early biological treatment within 18 months of diagnosis can modify the natural course of CD, minimize bowel damage, prevent disability, and improve clinical outcomes [11]. Recently, the PRedicting Outcomes For Crohn’s disease using a moLecular biomarker (PROFILE) trial highlighted the advantages of early combination therapy with infliximab and immunosuppressants [12]. This trial enrolled biologic-naïve CD patients within 6 months of diagnosis, randomizing them to either a top-down therapy group or an accelerated step-up therapy group. After one year of follow-up, the top-down group demonstrated a significantly higher rate of sustained steroid-free and surgical-free remission (79% vs. 15%). The findings of PROFILE suggest that top-down therapy with infliximab and immunosuppressants may be considered a standard approach for newly diagnosed CD patients.

While infliximab was the first biologic approved for CD, the availability of advanced therapies for treating CD continues to expand. It remains unclear whether early top-down treatment using biologics other than infliximab provides similar benefits. Ustekinumab (UST), a monoclonal antibody targeting the p40 subunit of interleukin-12 and interleukin-23, is recommended by the ECCO guidelines for inducing and maintaining remission in moderate-to-severe CD [9]. Additionally, real-world studies have shown that UST, when used as a first-line biologic, is as effective as other biologic agents in achieving clinical remission [13,14,15]. However, the impact of early UST treatment in biologic-naïve CD within six months of diagnosis remains unknown [16]. To date, no study has specifically evaluated whether early UST therapy improves clinical outcomes in CD. Therefore, this study aimed to address this critical gap.

## 2. Materials and Methods

### 2.1. Patients and Data Collection

In this multicenter retrospective cohort study, we initially enrolled 140 adult patients with CD who received UST between October 2020 and January 2024 at four medical centers in Taiwan: Chang Gung Memorial Hospital, Cheng Kung University Hospital, Tri-Service General Hospital, and Far Eastern Memorial Hospital. Patients were excluded if they had prior exposure to biologics or a follow-up duration of less than 26 weeks. Eligible patients were categorized into two groups based on the duration of CD at the start of UST treatment. Patients who initiated UST therapy within six months of CD diagnosis were placed in the Early-UST group, while those who initially received conventional therapies such as 5-ASA, steroids, and immunomodulators during the first six months and started UST treatment at a later stage were assigned to the control group.

We extracted data from the electronic medical records of each institution, collecting information on age, gender, smoking status, body mass index (BMI), time from diagnosis to UST initiation, Crohn’s Disease Activity Index (CDAI), Harvey–Bradshaw Index (HBI), previous intestinal resections, disease location and behavior (according to Montreal classification), IBD medication, laboratory data, and extraintestinal manifestations (EIMs). Data were collected at the time of UST initiation, as well as at 26 weeks (six months) and 52 weeks (one year) after UST initiation.

### 2.2. Clinical Outcome Evaluation

The UST treatment regimen comprised an initial intravenous (IV) induction dose at week 0, administered based on patient weight: 260 mg for those weighing ≤55 kg, 390 mg for those weighing between 55 and 85 kg, and 520 mg for those weighing ≥85 kg. Following the induction, patients received subcutaneous (SC) maintenance doses of 90 mg at week 8, with subsequent doses administered every 12 weeks. Follow-up assessments were performed at 26 weeks (6 months) and 52 weeks (1 year) after UST initiation.

The primary endpoint was the clinical outcomes six months after the first dose of UST, while the secondary endpoint was the outcome at one year. Clinical outcomes included CDAI scores, HBI scores, clinical remission rate, steroid-free clinical remission rate, BMI, hemoglobin, albumin, and C-reactive protein (CRP) levels. Clinical remission was defined as a CDAI score of less than 150, and steroid-free remission was defined as achieving clinical remission without concurrent steroid use. Adverse events were recorded throughout the follow-up period, with assessments conducted at six months and one year after UST initiation.

### 2.3. Statistical Analysis

Continuous variables were expressed as mean ± standard deviation, while categorical variables were presented as percentages. Student’s *t*-test was used to analyze continuous data, while the chi-square test or Fisher’s exact test was applied for categorical data. A *p*-value of <0.05 was considered statistically significant. All statistical analyses were conducted using SPSS version 29.0 (Armonk, NY, USA, IBM Corp).

## 3. Results

### 3.1. Baseline Characteristics

Initially, 140 patients with CD receiving UST treatment were assessed. Of these, 80 patients were excluded: 79 due to prior biologic exposure, and 1 due to a follow-up duration of less than six months. Ultimately, a total of 60 patients were included in the study. Among these, 24 patients (40%) initiated UST treatment within six months of diagnosis and were assigned to the Early-UST group, while the remaining 36 patients (60%) were classified as the control group. The patient inclusion process is illustrated in the flowchart (Figure 1).

The mean time interval from diagnosis to the first UST dose was 13.2 weeks in the Early-UST group, compared to 314.2 weeks in the control group (*p* < 0.001). Baseline characteristics, including age, gender, and smoking status, were comparable between the Early-UST group and the control group. The detailed baseline characteristics of the two groups are presented in Table 1.

### 3.2. Clinical Outcomes at Six Months

At the six-month follow-up, the Early-UST group demonstrated significantly lower CDAI scores (73.03 vs. 112.42, *p* = 0.038) and lower HBI scores (1.46 vs. 2.72, *p* = 0.020) compared to the control group. Additionally, a higher proportion of patients in the Early-UST group achieved clinical remission (91.7% vs. 63.9%, *p* = 0.017) and steroid-free remission (79.2% vs. 50.0%, *p* = 0.031). Laboratory parameters were comparable between the two groups.

Regarding adverse events, *Clostridioides difficile* infection was reported in one patient from the Early-UST group and two patients from the control group. Additionally, two patients in the Early-UST group and three patients in the control group required dose escalation at six months, which involved shortening the dosing interval from 12 weeks to 8 weeks. Hospitalizations and emergency department visits were similar between the two groups. No cases of malignancy or mortality were reported during the follow-up period. Additional details are provided in Table 2.

### 3.3. Clinical Outcomes at One Year

For the one-year outcome analysis, we excluded patients without complete one-year follow-up data. Specifically, seven patients from the Early-UST group and six from the control group were excluded due to a follow-up period of less than one year. Additionally, one patient in the control group was excluded due to mortality at 10 months. This left 17 patients in the Early-UST group and 29 patients in the control group available for the one-year clinical outcome assessment.

At one year, the Early-UST group maintained significantly lower CDAI scores (39.94 vs. 91.48, *p* = 0.005) compared to the control group (Table 2). Disease recurrence occurred in one patient from the Early-UST group and three from the control group. Among those who achieved clinical remission at six months, sustained remission was maintained in 93.8% of the Early-UST group and 87.5% of the control group. However, there were no significant differences between the two groups in terms of HBI score, remission rates, laboratory data.

As for adverse events, *Clostridioides difficile* infection occurred in two patients from the Early-UST group by one year, while *Clostridium innocuum* infection was observed in two patients from the control group. The single mortality case in the control group resulted from pneumonia complicated by septic shock and was not considered related to a CD-related opportunistic infection. By one year, dose escalation was required in three patients from the Early-UST group and six from the control group. The overall incidence of adverse events, including emergency room visits, hospitalizations, and opportunistic infections, was comparable between the two groups. Further details are illustrated in Table 2. There were minimal missing data, primarily in a few baseline variables such as BMI. The primary outcome variables had no missing data.

## 4. Discussion

In this multicenter, retrospective cohort study conducted in Taiwan, we found that the Early-UST group had significantly lower CDAI scores (73.03 vs. 112.42, *p* = 0.038), lower HBI scores (1.46 vs. 2.72, *p* = 0.020), higher rates of clinical remission (91.7% vs. 63.9%, *p* = 0.017), and higher rates of steroid-free remission (79.2% vs. 50.0%, *p* = 0.031) at six months compared to the control group. At one year, the Early-UST group continued to exhibit significantly lower CDAI scores (39.94 vs. 91.48, *p* = 0.005). These findings suggest that early UST treatment within six months of CD diagnosis can lead to a lower disease burden and better clinical and patient-reported outcomes. This is consistent with the PROFILE trial, which highlighted the importance of early biologic intervention in optimizing clinical outcomes. While the PROFILE trial demonstrated significantly higher rates of sustained steroid-free and surgery-free remission at one year in the early infliximab with immunosuppressant group [12], our study also showed significantly higher rates of clinical remission in the Early-UST group. Notably, none of the patients in our study required surgery during the one-year follow-up period.

Although no prior studies have specifically examined early UST use, the benefits of early use of anti-TNF-alpha agents were observed in subgroup analyses of the CHARM and PRECiSE2 studies [7,17]. Additionally, a cohort study by Faleck et al. demonstrated significantly higher rates of steroid-free remission (43% vs. 14%) at six months in CD patients who received earlier vedolizumab treatment (≤2 years duration) compared to those with later intervention (>2 years) [18].

Recent randomized controlled trials and real-world studies on biologic-naïve CD patients have reported six-month clinical remission rates with UST ranging from 54% to 67.7% when using an eight-week dosing interval [19,20,21]. In comparison, our study demonstrated six-month clinical remission rates of 91.7% in the Early-UST group and 63.9% in the control group with a twelve-week dosing interval, highlighting the importance of therapeutic timing for UST treatment. To explore potential factors influencing treatment response, we compared patients who achieved clinical remission with those who did not. The two groups were similar in age, gender, and Montreal classification. Additionally, we did not identify any statistically significant predictors for dose escalation in this study.

In terms of adverse events, UST, an IgG1 monoclonal antibody that binds to the p40 subunit shared by the pro-inflammatory interleukins 12 and 23, demonstrated an outstanding safety profile [22,23,24]. Unlike in other countries, in Taiwan, the National Health Insurance regulations stipulate a 12-week dosing interval for UST in IBD patients. Consequently, dose escalation (shortening the dosing interval) was required in 8.3% of patients during the first six months and in 19.6% of patients over the one-year period. Notably, apart from one 88-year-old patient with multiple comorbidities in the control group who died from aspiration pneumonia-related septic shock at ten months, there were no other treatment discontinuations throughout the twelve-month follow-up period in this study. Moreover, the incidence of adverse events, including opportunistic infections, hospitalizations, and emergency room visits, was comparable between the Early-UST group and the control group. These findings suggest that early UST treatment does not increase the risk of adverse events and leads to better clinical outcomes in newly diagnosed CD.

This study has several limitations. First, 40% of patients were using immunosuppressants at baseline, 11.7% at six months, and 21.7% at one year. Although immunosuppressants can be effective for induction and maintenance therapy [25], the proportion of patients receiving combination therapy did not differ significantly between the two groups at baseline or during outcome assessments. Additionally, previous studies have shown that concurrent use of immunosuppressants does not affect the rates of clinical remission and steroid-free remission in UST treatment for CD [26,27]. Second, the study is limited by its small sample size, relatively short follow-up period, and lack of data on fecal calprotectin, endoscopic remission, and transmural healing. These limitations are largely attributed to the reimbursement policies of Taiwan’s National Health Insurance, which restrict certain diagnostic tests, and the inherent constraints associated with a retrospective study design. Despite these limitations, this study is, to our knowledge, the first to evaluate the benefit of early UST treatment within six months of diagnosis in biologic-naïve CD patients. Future large-scale, long-term, prospective randomized controlled trials are warranted to confirm these findings.

In conclusion, initiating UST treatment within six months of diagnosis improves clinical and patient-reported outcomes in biologic-naïve CD patients. Therefore, it can be considered a superior therapeutic strategy in newly diagnosed CD patients.

## Figures and Tables

**Figure 1 biomedicines-13-00391-f001:**
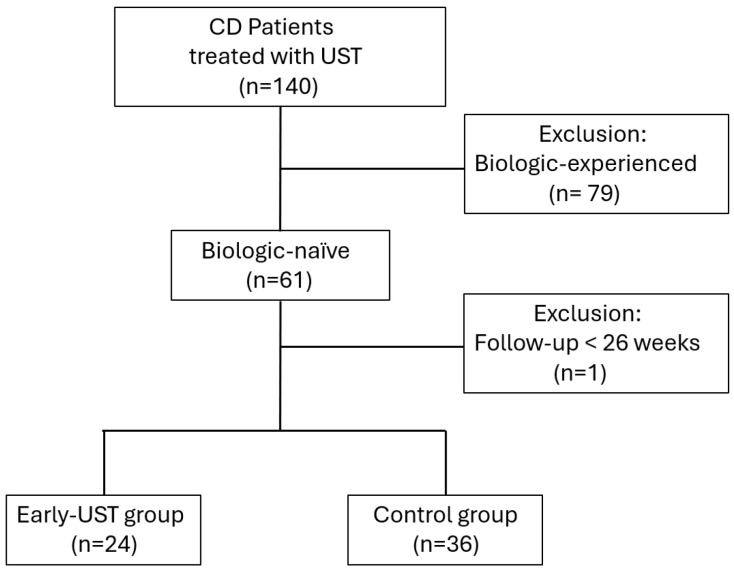
Flowchart of patient selection and grouping in this study. This flowchart illustrates the process of screening and categorizing patients with Crohn’s disease (CD) who received ustekinumab (UST) treatment. It includes the initial number of patients assessed, the exclusion criteria applied, and the final grouping into the Early-UST group and control group.

**Table 1 biomedicines-13-00391-t001:** Baseline characteristics of biologic-naive Crohn’s disease patients receiving ustekinumab treatment in Early-UST and control groups.

	Overall	Early-UST	Control (n = 36)	*p*-Value
	(n = 60)	(n = 24)		
Baseline Characteristics				
Age at biologics initiation (years)	40.53 ± 17.53	36.63 ± 12.73	43.14 ± 19.85	0.127
Gender, male (%)	45 (75.0)	16 (66.7)	29 (80.6)	0.224
Smoker (%)	11 (18.3)	3 (12.5)	8 (22.2)	0.34
BMI	21.67 ± 4.90	22.21 ± 4.40	21.31 ± 5.23	0.487
Time from diagnosis to UST initiation (weeks)	193.77 ± 326.73	13.19 ± 7.51	314.16 ± 377.69	<0.001 *
CDAI score	284.53 ± 74.21	266.81 ± 86.72	296.34 ± 63.10	0.16
HBI score	6.50 ± 2.83	6.42 ± 2.92	6.49 ± 2.87	0.913
Previous intestinal resection (%)	14 (23.3)	4 (16.7)	10 (27.8)	0.319
Montreal classification (%)				
L1, ileal	25 (41.7)	11 (45.8)	14 (38.9)	0.830
L2, colonic	6 (10.0)	1 (4.2)	5 (13.9)	0.387
L3, ileocolonic	29 (48.3)	12 (50.0)	17 (47.2)	0.792
L4, isolated upper disease	0	0	0	-
B1, non-stricturing, non-penetrating	39 (65.0)	14 (58.3)	25 (69.4)	0.418
B2, stricturing	13 (21.7)	8 (33.3)	5 (13.9)	0.110
B3, penetrating	8 (13.3)	2 (8.3)	6 (16.7)	0.457
P, perianal disease	11 (18.3)	3 (12.5)	8 (22.2)	0.500
Concomitant IBD medication (%)				
Steroid	43 (71.7)	18 (75.0)	25 (69.4)	0.640
Immunosuppressant	24 (40.0)	7 (29.2)	17 (47.2)	0.162
5-ASA	34 (56.7)	14 (58.3)	20 (55.6)	0.832
Lab data				
Hemoglobin (g/dL)	12.52 ± 2.39	13.05 ± 1.69	12.17 ± 2.72	0.128
Albumin (g/dL)	4.01 ± 0.68	4.18 ± 0.37	3.93 ± 0.80	0.747
CRP (mg/L)	14.29 ± 31.78	11.67 ± 19.16	16.03 ± 38.12	0.562
EIM	8 (13.3)	3 (12.5)	5 (13.9)	1.000

BMI, body mass index; CDAI, Crohn Disease Activity Index; CRP, C-reactive protein; EIM, extraintestinal manifestation; HBI, Harvey–Bradshaw Index; IBD, inflammatory bowel disease; n, number; UST, ustekinumab. * *p* < 0.05.

**Table 2 biomedicines-13-00391-t002:** Clinical outcomes in biologic-naive Crohn’s disease patients receiving ustekinumab treatment in Early-UST and control groups.

	6 Month				1 Year			
	Overall (%)	Early-UST (%)	Control (%)	*p*-Value	Overall (%)	Early-UST (%)	Control (%)	*p*-Value
	(n = 60)	(n = 24)	(n = 36)		(n = 46)	(n = 17)	(n = 29)	
CDAI score	96.66 ± 72.41	73.03 ± 65.90	112.42 ± 73.12	0.038 *	72.01 ± 68.99	39.94 ± 43.13	91.48 ± 74.92	0.005 *
HBI score	2.22 ± 2.08	1.46 ± 1.69	2.72 ± 2.17	0.020 *	1.37 ± 1.55	0.88 ± 1.27	1.66 ± 1.65	0.104
Concomitant IBD medication (%)								
Steroid	9 (15.0)	3 (12.5)	6 (16.7)	0.729	10 (21.7)	3 (17.6)	7 (24.1)	0.723
Immunosuppressant	7 (11.7)	2 (8.3)	5 (13.9)	0.691	10 (21.7)	2 (11.8)	8 (27.6)	0.282
Clinical remission	45 (75.0)	22 (91.7)	23 (63.9)	0.017 *	36 (78.3)	16 (94.1)	21 (72.4)	0.124
Steroid-free remission	37 (61.7)	19 (79.2)	18 (50.0)	0.031 *	28 (60.9)	13 (76.5)	15 (51.7)	0.205
Lab data								
Hemoglobin	13.04 ± 2.28	13.51 ± 1.84	12.72 ± 2.52	0.272	13.30 ± 2.30	13.51 ± 1.84	12.98 ± 2.68	0.507
Albumin	4.28 ± 0.51	4.39 ± 0.28	4.20 ± 0.26	0.841	4.28 ± 0.52	4.39 ± 0.23	4.22 ± 0.62	0.788
CRP	7.55 ± 14.43	3.52 ± 5.58	10.04 ± 17.46	0.585	7.41 ± 25.56	1.37 ± 1.21	11.35 ± 32.49	0.136
BMI	22.84 ± 4.89	23.08 ± 5.67	22.68 ± 4.34	0.790	22.44 ± 4.01	23.58 ± 6.19	22.42 ± 5.17	0.870
Adverse event (%)								
Malignancy	0	0	0	-	0	0	0	-
Opportunistic infection	3 (5.0)	1 (4.2)	2 (5.6)	1.000	6 (13.0)	2 (11.8)	4 (13.8)	1.000
CD-related ED visits (%)	4 (6.7)	0	4 (11.1)	0.121	4 (8.7)	1 (5.9)	3 (10.3)	0.619
CD-related hospitalization (%)	4 (6.7)	1 (4.2)	3 (8.3)	0.632	5 (10.9)	2 (11.8)	3 (10.3)	1.000
Dose escalation	5 (8.3)	2 (8.3)	3 (8.3)	1.000	9 (19.6)	3 (17.6)	6 (20.7)	1.000
EIM (%)	2 (3.3)	0	2 (5.6)	0.512	3 (6.5)	2 (11.8)	1 (3.4)	0.547

BMI, body mass index; CD, Crohn disease; CDAI, Crohn Disease Activity Index; CRP, C-reactive protein; EIM, extraintestinal manifestation; ED, emergency department; HBI, Harvey–Bradshaw Index; IBD, inflammatory bowel disease; n, number; UST, ustekinumab. * *p* < 0.05.

## Data Availability

The datasets analyzed in the study are available from the corresponding author upon reasonable request due to privacy and ethical reasons.

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
