# Peer review of "Early Ustekinumab Use Improves Clinical Outcomes in Biologic-Naive Crohn’s Disease Patients: A Retrospective Multicenter Cohort Study in Taiwan"

_biomedicines, 2025, doi:10.3390/biomedicines13020391_

Round 1
Reviewer 1 Report
Comments and Suggestions for Authors
This study demonstrates that early introduction of Ustekinumab improves disease outcomes compared to later initiation of biologics. Overall, the study is well-designed, particularly in the context of a relatively rare population for Crohn's disease. However, there are a few points that need clarification:
-
Throughout the study, the authors used the CDAI score and HBI score to define remission. Can the authors confirm whether these patients also achieved endoscopic remission or even histological remission after 6 months of Ustekinumab treatment?
-
It is not clear whether patients who achieved remission at 6 months still maintained remission after 1 year of follow-up. How many of the early Ustekinumab patients had disease recurrence at the 1-year follow-up?
-
Do we have information on the phenotypic characteristics of patients who did not respond to treatment?
-
For biologic-experienced patients, is there any evidence to suggest that starting Ustekinumab immediately after therapeutic failure leads to a better response compared to later introduction of biologics?
Reviewer 2 Report
Comments and Suggestions for Authors
Dear editor,
I appreciate the opportunity to review the manuscript titled "Early Ustekinumab Use Improves Clinical Outcomes in Biologic-Naive Crohn’s Disease Patients: A Retrospective Multicenter Cohort Study in Taiwan"
This study addresses an important question regarding the timing of ustekinumab (UST) initiation in biologic-naïve Crohn’s disease (CD) patients. The findings, which suggest improved clinical outcomes with early UST use, are relevant and contribute to the ongoing exploration of optimal therapeutic strategies for CD. However, several methodological and interpretational issues need to be addressed before publication.
Abstract
Clarify the definition of the “control group.” Indicate whether these patients received UST at a later stage or were treated with other therapies.
Material and Methods
· The number of enrolled and excluded patients is mentioned inconsistently. For instance, it’s unclear why the authors refer to a “control group” when those patients also received UST. Clarifying this point is crucial for the validity of the comparison.
· The authors should specify the induction (IV) and maintenance (SC) regimens of UST clearly, as the current description is somewhat ambiguous.
· In line 78, the authors said, " we enrolled adult patients with CD", the number of enrolled patients should be stated.
· Consider moving the description of the UST dosing regimen at the beginning of sub-heading "2.2. Clinical outcomes evaluation". Then describe the primary and secondary endpoint.
· Explain the significance of the Taiwanese healthcare system’s 12-week dosing interval policy and how it may influence the study outcomes compared to other global settings that consider 8-weeks dosing interval.
Results
· In line 118, the authors state: "Of these, 80 patients were excluded due to prior biologic exposure (n=79) or a follow-up duration of less than six months (n=1)." This sentence should be rephrased for clarity and grammatical accuracy. A more appropriate version would be: "Of these, 80 patients were excluded; 79 due to prior biologic exposure, and one due to a follow-up duration of less than six months.
· Clarify why some patients failed to achieve clinical remission or required dose escalation. Was this due to disease severity, pharmacokinetics, or other factors?
· The absence of long-term follow-up data (beyond one year) limits the generalizability of the findings.
· Expand the adverse event section with greater detail on infections and dose escalation, particularly since the control group included one death.
Discussion
· The discussion should begin with the study’s key findings rather than a reiteration of background information.
· The authors do not sufficiently explore why some patients failed to achieve remission or experienced dose escalation. A more detailed analysis of these cases would improve the discussion.
· In lines 195-200, the authors claimed that early UST is superior needs to be balanced with a comparison to existing treatments, such as infliximab, especially since infliximab demonstrated statistically significant outcomes in prior studies.
· Move the discussion on UST’s therapeutic potential (lines 172-187) to the introduction for better flow.
Minor comments
· Ensure consistent terminology, replacing “anti-TNF” with “anti-TNF-alpha.”
· The “Declaration” heading should replace “Patents” in the final section.
· Remove redundant phrases and ensure concise, focused statements.
· Grammatical issues throughout the text should be addressed.
Comments on the Quality of English Language
Its Ok.
